# Changing Human Behavior to Improve Animal Welfare: A Longitudinal Investigation of Training Laboratory Animal Personnel about Heterospecific Play or “Rat Tickling”

**DOI:** 10.3390/ani10081435

**Published:** 2020-08-17

**Authors:** Megan R. LaFollette, Sylvie Cloutier, Colleen M. Brady, Marguerite E. O’Haire, Brianna N. Gaskill

**Affiliations:** 1The North American 3Rs Collaborative, 251 Ballardvale, Wilmington, MA 01887, USA; 2Department of Animal Sciences, College of Agriculture, Purdue University, West Lafayette, IN 47907, USA; bgaskill@purdue.edu; 3Independent Scientist, Ottawa, ON, Canada; scloutier@ccac.ca; 4Department of Agricultural Sciences Education & Communication, College of Agriculture, Purdue University, West Lafayette, IN 47907, USA; bradyc@purdue.edu; 5Center for the Human-Animal Bond, Department of Comparative Pathobiology, College of Veterinary Medicine, Purdue University, West Lafayette, IN 47907, USA; mohaire@purdue.edu

**Keywords:** animal welfare, rat tickling, laboratory rats, human-animal interaction, heterospecific play, theory of planned behavior, training, handling, playful handling, behavior change

## Abstract

**Simple Summary:**

When laboratory rats are first handled, they can experience fear and stress, which negatively influences animal welfare. Rat tickling, a positive handling technique, can improve these outcomes. However, despite evidence for rat tickling’s animal welfare benefits, the technique is rarely implemented, in part because of a lack of training. Our purpose was to determine the effectiveness of two rat tickling training programs (as compared to a control treatment) on reported implementation, self-efficacy, knowledge, familiarity, and beliefs about rat tickling. After completing an initial survey, 96 laboratory animal personnel currently working with rats were assigned to receive online-only training, online + hands-on training, or no training (control condition). Participants received further surveys directly after training and 2 months later. Results showed that both online-only and online + hands-on training improved key outcomes for rat tickling (i.e., increased implementation, self-efficacy, knowledge, and familiarity rat tickling). Online + hands-on training had a few additional benefits (i.e., increased control beliefs and greater increases for self-efficacy and familiarity with rat tickling). Overall, these findings support the development of targeted interactive training programs to improve the implementation of potential welfare-enhancing techniques.

**Abstract:**

Despite evidence for rat tickling’s animal welfare benefits, the technique is rarely implemented in part because of a lack of training. This study’s purpose was to determine the efficacy of online-only or online + hands-on training programs on key outcomes for rat tickling in comparison to a waitlist control condition. After completing a baseline survey, laboratory animal personnel currently working with rats in the United States were semi-randomized to receive online-only training (*n* = 30), online + hands-on training (*n* = 34), or waitlist control (*n* = 32). Participants received further surveys directly after training and 2 months later. Data were analyzed using general linear mixed models. At the 2-month follow-up compared to baseline, both training groups reported increased implementation, self-efficacy, knowledge, and familiarity of rat tickling while only the online + hands-on training participants reported increased control beliefs (while the waitlist group stayed the same). At the 2-month follow-up compared to the waitlist, hands-on training participants reported increased self-efficacy and familiarity with rat tickling. Overall, findings show that both online-only and online + hands-on training can improve key outcomes for rat tickling. Although online + hands-on training is slightly more effective, the interactive online-only training has the potential to improve widescale implementation of a welfare-enhancing technique.

## 1. Introduction

Although scientific research has provided a variety of well-supported strategies to improve animal welfare across species and settings, these findings are often not widely implemented. Examples of issues where a lack of implementation has been reported despite known strategies include pig aggression [1], lameness in dairy cows [2], and laboratory rodent handling [3]. This lack of implementation should be of concern to researchers and funding agencies who wish to see scientific results translated into everyday practices to improve animal welfare. If research findings are not translated to practice, then their applied benefits are unrealized.

One example of these circumstances can be found in laboratory rats. These animals can perceive handling negatively, especially when naive to human interactions, which can result in fear, stress, anxiety, and even more difficult handling [4,5,6]. During initial handling, especially if an intentional effort is not made, negative effects can be further increased by common laboratory procedures, such as marking the animals for identification, restraint, injection, and blood draws [7]. Beyond negatively impacting rat welfare, stress is also a potential confounding factor for scientific experiments [8]. Therefore, the negative effects of stress during handling both reduce the possible benefits that can be gained from scientific research (i.e., by reducing study validity and reliability) and increase the costs of scientific research (i.e., by harming rat welfare).

A refinement to rat research and improvement to rat welfare can be made through the use of the positive handling technique heterospecific play, or “rat tickling”. This technique mimics aspects of rat rough-and-tumble play [9]. A systematic review of 53 experiments in rat tickling shows that this technique increases rat positive affect, habituation to handling, and positive approach behaviors, thus reducing stress associated with routine handling [10]. Rat tickling can even reduce or eliminate negative responses to repeated intraperitoneal injections when administered just prior to the injection [11].

Despite the strong scientific evidence of its benefits, our 2018 survey of almost 800 laboratory animal personnel indicated that 55% of participants never use rat tickling [3]. However, personnel’s frequency of use of rat tickling was strongly associated with their beliefs about and familiarity with the technique. For example, participants were more likely to provide rat tickling if they more strongly believed that rat tickling was beneficial, expected by their peers, and that they were confident in their ability to implement rat tickling. When participants were asked in a free response question to identify what made it difficult to provide rat tickling, the most common barrier identified was a lack of time (stated by 60% of participants). However, this barrier is addressed by recent research showing that only 15 s of tickling per day for 3 days is sufficient to elicit positive responses from the rats [12]. Other commonly listed barriers were personnel buy-in and a lack of training.

Currently, the causal link is missing between training in rat tickling (or other enrichment techniques) and important outcomes related to implementation. However, in the field of farm animal welfare, research shows that targeted in-person training can effectively improve stockperson beliefs and increase the implementation of positive handling techniques [13]. Furthermore, experts recommend that welfare findings should be communicated outside of the primary research community, especially with key initial stakeholders, and under close supervision to ensure success [14]. Despite this, to our knowledge, a study has not been conducted on training laboratory personnel in the research environment (which has considerable more variation than large-scale animal agriculture, which typically houses a single species for a singular purpose with homogeneous housing, such as egg-laying hens in an aviary system).

Moreover, it remains unclear what training modality is necessary to implement change. Laboratory personnel self-report that hands-on courses are influential to learning handling techniques [15]. However, it may not be feasible to rapidly disseminate new information about welfare-enhancing techniques through hands-on workshops alone, due to time and financial limitations. Furthermore, even if hands-on workshops are available, providing background material online before participants attend the training would save instructors significant amounts of time. Online education is advantageous since it can efficiently reach a larger number of participants with minimal costs. However, its efficacy in teaching hands-on procedures is unknown. Based on the current stress of rats in response to handling, but lack of implementation of the effective technique of rat tickling, there is a critical need to assess the impact of rat tickling training materials, in both online-only and online/hands-on formats, on laboratory animal personnel attitudes, knowledge, and implementation.

This study’s objective was to conduct a methodologically rigorous longitudinal trial to quantify the efficacy of training programs on laboratory animal personnel, with the end goal of improving animal welfare. Our specific aims were to quantify the efficacy of different training programs of laboratory personnel beliefs, self-efficacy, knowledge, and implementation of rat tickling. Based on previous research using the theory of planned behavior and personal experience, we hypothesized that, relative to a waitlist control, laboratory personnel who undergo training programs would report improved attitudes, self-efficacy/knowledge, and implementation of rat tickling. We further hypothesized that there would be an additive benefit to hands-on training relative to online-only training. With this knowledge, we hoped to experimentally identify effective training programs to increase rat tickling prevalence as well as establish a model for improving the implementation of other animal welfare-enhancing techniques.

## 2. Materials and Methods

All procedures and informed consent protocols were approved by Purdue University’s Human Research Protection Program Institutional Review Board (IRB), protocol #1712020004. All interactions between researchers, study participants, and rats during the study were approved by each individual university’s Institutional Animal Care and use Committee (IACUC): Harvard University protocol #14-02-189-1, University of California San Francisco protocol #AN180239-01B, Indiana University Purdue University Indianapolis protocol #11426, and Purdue University protocol #1201000547.

### 2.1. Participants and Procedures

Participants were recruited between August 21nd and September 4th, 2019 via widespread online promotions designed to maximize sample size [16]. Online contacts were through seven modalities: Direct emails to known laboratory personnel, emails to individuals at each host institution, list serves (e.g., CompMed, Laboratory Animal Research Enrichment Forum (LAREF), etc.), email lists (e.g., MSMR), Facebook groups/pages/personal accounts (e.g., Laboratory Animal Sciences), and LinkedIn groups/personal pages (e.g., American Association for Laboratory Animal Science (AALAS), Animal Behavioral Biology). All modalities were contacted up to three times with the same study flyer following recommended survey procedures [17].

Participants were included if they were over the age of 18 and reported that they had worked with laboratory rats in the last 12 months or planned to work with laboratory rats between August and December 2019 in the United States. We informed participants that work was defined broadly to include both hands-on (e.g., changing cages, performing procedures) and hands-off work (e.g., supervision as a clinical veterinarian, principle investigator, or manager). To compensate participants for their time, they received one entry into a drawing for a choice between a $40 Amazon e-gift card or cash (chosen by 76% and 22%, respectively, with one participant donating their prize back to the research team to use for future research) each time they completed a survey (50 prizes available).

### 2.2. Experimental Treatments

Figure 1 details the treatment group assignment and a timeline of procedures. Three treatment groups were evaluated in this study. The online-only training group received an online training course about rat tickling. The online + hands-on training group received the same online training course as well as a small group, in-person, hands-on training session. The waitlist control group received no interventions during the study period but were given access to the online training course after completing the final survey (and told in advance that they would get this access).

All treatment groups were evaluated at three time points. First, all participants completed a baseline survey before being assigned to treatment group (baseline). Second, participants completed a survey directly after completing their assigned training (i.e., the online training course for the online-only group or the hands-on training for the online + hands-on group; post-training). The waitlist received a second survey at a similar time point. Third, participants completed a final survey approximately two months after the post-training survey (2 months).

A partially randomized controlled trial design was used in this project in order to maximize sample sizes in each group. Although a completely randomization design was intended, not enough participants were recruited that were available on the hands-on training session days. Therefore, after completing the baseline survey (in which they indicated their availability for a hands-on training session), participants were assigned to one of three treatment groups. If they were available to participate in the hands-on training, they were assigned to that group. Otherwise, they were randomized using a random number generator (random.org) to either the online-only or waitlist control treatment groups. Additionally, 4 individuals were originally assigned to the online + hands-on training group and took the online course in preparation, but then were unable to attend the workshop; these individuals were re-assigned to the online training group.

Participants in the online-only training group were given a link to the post-training survey upon course completion to ensure they had truly completed the course. Participants in the online + hands-on group were required to send a screenshot of their course completion certificate before they were allowed to participate in the hands-on training. After completing the hands-on training, participants filled out a paper version of the post-training survey.

#### 2.2.1. Online Training Course

Prior to this study, our research team had conducted 7 hands-on workshops in the United States and Canada over the previous 3 years. Participants in these prior hands-on workshops included diverse perspectives from a variety of roles (students, animal caretakers, veterinarians, researchers, managers), research fields (neuroscience, behavioral, training, basic), and institutions (industry, academic). These workshops included both informational lecture and practical hands-on sections. During these prior workshops, we identified and addressed common difficulties with the hands-on procedures, misconceptions, and frequently asked questions. During three years of teaching, we refined our methods and explanations of teaching individuals the tickling technique. For example, we have repeatedly found that the transition from dorsal contact to pin is the most difficult part of rat tickling for participants. Therefore, we added extensive pictorial, video, and hands-on instruction on this technique and clarified a very specific recommended method for this transition that we have found to be the most effective for most participants.

The online training course for this study was carefully and intentionally developed from our previous training experience and research. The online course was designed using Articulate Storyline™ software to create a seamless course complete with multiple interactive elements as well as extensive video and pictorial examples of rat tickling. Dick and Carey’s systems approach to instructional design was used [18] and took an average of 30 min to complete. Course topics included the rationale behind rat tickling, detailed pictorial/video instruction on the hands-on technique (including videos of both what to do and what not to do), guidance on implementation, how to assess a rat’s response to the technique (including videos of positive and negative rat reactions to tickling) and respond appropriately, and a series of frequently asked questions. Assessing a rat’s response to the technique is particularly important since rats show a wide range of individual differences in response to rat tickling [10]. Based on previous literature, we recommend tickling juvenile rats for at least 3 days [12] before assessing, unless overt aggression or extreme distress is seen. This allows rats to learn about the interaction, habituate, and benefit from it. Additionally, we cautioned participants against tickling in older rats especially without extensive prior handling experience, highly stressed rats, and breeder rats. Multiple videos and pictures from a variety of angles were used to communicate the hands-on technique as clearly as possible. In order to advance to the next section within the course, participants had to complete a quiz about that section’s content. Slides were carefully designed to engage participants and communicate in a clear manner. The most updated version of this course can be found at: bit.ly/RatTicklingCertificate.

#### 2.2.2. Hands-On Training Session

The hands-on training workshop was focused on teaching hands-on skills, rather than the theory behind rat tickling. Each session had a maximum of 5 participants and lasted approximately 45 min. Two instructors (MRL and BNG) led each session. Participants first observed the session leaders prior to attempting the technique. Participants were given immediate individualized feedback on their hands-on technique throughout the workshop. Session leaders also noted key rat behaviors during the workshop indicative of either positive or negative responses to tickling. Refinement techniques were used to first train participants with stuffed rats, then pre-tickled rats, and finally naïve rats (although to minimize rat numbers, not all participants were able to work with completely naïve rats). This order of interaction also allowed participants to work with rats of increasing handling difficulty. On the first day of tickling, it is not unusual for rats to produce fewer 50-kHz vocalizations and be somewhat resistant to an entirely novel interaction with a human. However, after 3 tickling sessions, most juvenile rats habituate and show behavioral signs of positive affect [12].

The rats used in training sessions varied due to host institutional needs, participant sign-ups, and to apply the 3Rs principle of reduction to minimize additional rats that would not find subsequent use. At the two training locations that could find subsequent use for new rats, 20 juvenile 6-week-old Long-Evans rats (Crl:LE) were ordered for the workshop. At the other two locations, we used available surplus rats from studies or breeding colonies (i.e., 10 8-week-old Crl:CD rats, and 4 8-week-old and 4 12-week-old Crl:CD).

Bat detectors (Bat 4 Bat Detector, Magenta Electronics Ltd., Staffordshire, UK) set to 5 kHz were used during the hands-on sessions to allow participants to hear the positive ultrasonic vocalizations while tickling the rats. These devices transduce ultrasonic vocalizations at a very specific level into a range audible to humans in real time. Similar devices were used in the original rat tickling experiments [9] (Panksepp and Burgdorf 2000). Although these devices do not capture the full range of possible vocalizations (20–75 kHz), they are very simple to use and can be obtained at a significantly lower price range than other ultrasonic microphones. Therefore, we recommended to participants to purchase such detectors for their laboratories. Many participants noted that using a bat detector was extremely beneficial for them to get real-time feedback from the rats while learning.

### 2.3. Measures

This survey was developed by reviewing the literature and consulting with experts in survey methodology, behavior theory, and laboratory animal enrichment. When possible, validated instruments were used (i.e., theory of planned behavior survey). When validated instrumentation did not exist, previous work was modified or new items were created, reviewed by experts, piloted, and revised as necessary. Several measures were based off a cross-sectional survey of rat tickling in the laboratory [3]. The survey question text and scales are available in Appendix A. Unless otherwise noted, each question was given during every survey.

#### 2.3.1. Demographics and Work Factors

Participants were asked about their demographics and current work in the baseline survey only. Demographics included age, gender, race/ethnicity, and highest level of education. Current work questions included role (e.g., animal care technician, veterinarian), if they supervised others, type of institution (e.g., academic, contract research organization), primary type of research (e.g., applied, basic, regulatory), highest level of professional certification, and number of years working with laboratory animals.

#### 2.3.2. Baseline Factors

Baseline factors that could influence the uptake of the rat tickling procedure were measured only in the baseline survey. Participants were asked how many hours they work with rats in a typical work week, the degree of stress or pain most of their rats’ experience via the USDA pain and distress scale [19], and how confident they were in their general rat handling skills. Participants were also asked how much control they had over the provision of enrichment to their rats, if they wished they could provide more enrichment than they currently do, and how, if at all, they had previously heard about rat tickling (e.g., journal article, popular press article).

#### 2.3.3. Current Implementation and Future Intention to Tickle Rats

A few key questions were used to assess participants’ current and future implementation of rat tickling. Participants were asked how often they provided tickling to laboratory rats in the past 2 weeks (baseline and post-training surveys) or 2 months (follow-up survey). Answer options for this question included never (0% of instances working with rats), rarely, sometimes (50% of instances working with rats), often, or always (100% of instances working with rats). Participants were also asked about their intentions (e.g., want, expect, and intend) to provide rat tickling in the next year using protocols from the theory of planned behavior [20]. Individuals who supervised others were also asked these questions about those that they supervised.

In the final survey, to determine if individuals could identify the correct scientifically supported tickling technique, participants were asked to identify which picture most accurately represents the correct method for how laboratory rats should be tickled (Figure 2).

#### 2.3.4. Knowledge, Self-Efficacy, and Familiarity with Rat Tickling

Participants were asked several questions to determine their knowledge, self-efficacy, and familiarity with rat tickling. Self-efficacy to tickle rats was assessed via 5 questions modified from a general self-efficacy scale [21]. This scale asked participants about their confidence on their ability to tickle rats in general, naïve rats, and complete the three components of rat tickling: dorsal contact (“nape”), flip, and pin (“on belly”). These targeted self-efficacy questions were included since previous workshop participants have anecdotally reported that the flip and pin, arguably the most important components of rat tickling, are also the most difficult. Knowledge about rat tickling was assessed via 7 factual questions about the technique. Participants were also asked about when tickling should be implemented in relation to procedures and study timeline, duration, frequency, rationale, whether tickling or stroking is better, and whether adult rats should ever be tickled. Finally, familiarity was assessed via a single question asking participants “how familiar are you with rat tickling? Please think about both your general knowledge and hands-on technique.”

#### 2.3.5. Beliefs about Rat Tickling

Beliefs about rat tickling were assessed using a brief theory of planned behavior questionnaire based off our previous research [3]. Surveys constructed using this theory typically have excellent reliability and validity [22]. Participants answered 9 close-ended quantitative questions about their behavioral attitudes (consequences of rat tickling), subjective norms (social and professional pressures to provide rat tickling), and perceived behavioral control (general confidence/control over the ability to apply rat tickling). The perceived behavioral control variable is characteristically different from self-efficacy as it asks participants about external control factors, such as whether providing rat tickling is actually under the control of the participant (e.g., an animal caretaker may be confident in their ability to provide rat tickling but not allowed to provide it to their rats because of managerial decisions).

#### 2.3.6. General Human–Rat Interactions

Participants also received a general behavior survey adapted from Hemsworth and Coleman (20). Participants were asked if they agreed or disagreed that they often observe, pet, talk to, or name their laboratory animals.

#### 2.3.7. Qualitative Questions

At several points throughout the survey, participants were asked to answer open-ended qualitative questions. These questions allowed participants to reply with their most salient answers without additional prompting. During each survey, after asking about implementation, participants were asked if they had any further comments about their previous experiences with rats or rat tickling. Then, participants answered questions about rat tickling benefits (i.e., what are the advantages to rat tickling) and barriers (i.e., what makes it difficult for you to tickle rats). At the end of each survey, participants were asked if they had any final comments.

### 2.4. Data Analysis

#### 2.4.1. Participant Inclusion and Variable Coding

A total of 182 laboratory animal personnel began the baseline/screening survey and answered all 3 screening questions. Of those, 16% (*n* = 29) were excluded for not being located in the United States or not currently working with laboratory rats. Of those, only individuals that completed the baseline survey (80%, *n* = 122) were assigned or randomized to treatment groups and invited to participate in the post-training survey (Figure 1). Of those, 80% (*n* = 97) actually completed their treatment assignment and the post-training survey. Those that completed the second survey were then invited to complete the third survey, of which 90% (*n* = 87) actually completed the third survey. Of those individuals who completed the first two surveys, to ensure that all descriptive data reporting and summary scores indicate the same responses, only participants that answered at least 50% of the questions for each scale in the survey were included for analysis (ultimately, this only excluded 1 additional participant from the analysis).

#### 2.4.2. Quantitative Analysis

Data analysis was conducted in JMP Pro 14.0.0(SAS Institute GmbH, Heidelberg, Germany) using descriptive statistics, chi-squared tests, and general linear mixed models. Prior to testing, all assumptions of the general linear model were visually confirmed, including independence of residuals, homogeneity of variance, and normality of residuals. For all summary scales, an average of individual items was calculated (excluding participants missing over 50% of the data in each measure). Significant main effects and two-way interactions were analyzed using Tukey tests. A chi-squared test was used to analyze the dependent variable of correct identification of the rat tickling technique compared to the treatment group.

The dependent variables for quantitative analysis via general linear mixed models were rat tickling implementation (i.e., implementation, intent), knowledge, self-efficacy, familiarity, and beliefs (i.e., behavioral attitudes, subjective norms, and perceived behavioral control). The independent variables of interest were treatment (i.e., online-only training, online + hands-on training, waitlist control), time point (i.e., baseline, post-training, 2-month follow-up), and an interaction of treatment and time point. To control for potential confounding effects, we also included baseline variables of rat stress/pain level, enrichment factors (control and desire), and confidence in rat handling skills. These potential confounding effects were removed from the model if *p* > 0.05. To avoid pseudoreplication and accommodate repeated measures, analyses were blocked by the random experimental unit of participant with treatment nested within. The significance level was *p* < 0.05.

Overall, the exemplary initial analysis used was:

Dependent variable = treatment + time point + treatment × time point + rat stress/pain + enrichment control + enrichment desire + rat handling experience + participant (treatment).

#### 2.4.3. Qualitative Analysis

We used thematic content analysis to analyze responses to open-ended questions. We were interested in determining participant-identified benefits and barriers to tickling as well as specific comments about the online or hands-on trainings. An iterative process was used to code the entire qualitative data set. Within the dataset, each clause was treated as the unit of analysis and each clause given a code. Each clause was coded with as many codes as it contained. For example, the clause “time and buy in from management that it is a beneficial practice” would receive codes Time and Buy-In. Buy-in was defined as a belief that rat tickling is effective and worth the time and effort it requires. The coding manual was developed from our previous research coding 794 responses of laboratory animal personnel to similar questions [3]. New codes were added as needed to accurately describe this new dataset. The coding manual was refined via an interactive process in which responses were read multiple times.

## 3. Results

### 3.1. Demographics

A total of 96 participants completed at least the first two data collection points in the study and therefore were included in the study (30 = online-only, 34 = hands-on, 32 = waitlist). Additionally, 86 participants completed all 3 timepoints (28 = online-only, 28 = hands-on, 30 = waitlist). Detailed demographic information is displayed for all participants in Appendix A. Overall, participants were primarily white (79%) females (82%) with a bachelor’s degree or higher (79%). On average, they were 37 years old, worked with rats 7 h a week, and had been working in the laboratory animal field for 11 years. Participants worked mostly in universities (75%), but in a variety of roles (e.g., 16% managers, 22% laboratory technicians, 16% veterinarians) and research types (e.g., 48% applied). The majority had some sort of laboratory animal certification (86%). Less than half of participants (42%) currently supervised others working with rats. There were no significant group differences in demographics between treatment groups (Appendix A).

### 3.2. Baseline Characteristics

The majority of participants self-reported that most of their rats generally experience minor stress or pain of a short duration based on the United States Department of Agriculture (USDA) pain scale (55%). Although 93% of participants indicated they had at least a little control or influence over their rats’ enrichment, only 20% indicated that they had complete control. The majority of participants (76%) wished they could provide their rats more enrichment than they currently do. The majority of participants (90%) were also confident in their general rat handling skills. At baseline, more than a third of participants (38%) were either not at all or only slightly familiar with rat tickling, while 38% were somewhat familiar, and 25% were moderately to very familiar with rat tickling. Over half of participants (62%) had seen an educational talk about rat tickling previously. Participants had also heard of rat tickling through technical articles (49%), peer-reviewed journal articles (32%), and YouTube videos (33%).

At baseline, there were almost no differences in outcome measures or covariates between treatment groups (*p* > 0.05 for implementation, intent, familiarity, self-efficacy, knowledge, attitudes, norms, control beliefs, enrichment control, enrichment desire, and rat handling experience; Table 1). The only difference was the self-reported rat stress/pain based on the USDA pain categories. Higher rat stress/pain was reported by the hands-on group compared to the waitlist (Table 1).

### 3.3. Impacts of Treatment, Time, and Baseline Factors.

#### 3.3.1. Implementation, Intention, and Technique

Self-reported implementation of rat tickling was significantly impacted by the interaction of treatment and time (Figure 3a, Table 2). Compared to baseline, the online + hands-on training groups had higher implementation immediately after training and 2 months post-training (Tukey, *p* < 0.05), while the online training group only had higher implementation at 2 months post-training (Tukey, *p* < 0.05). Within each time point, no group was significantly different from the others (Tukey, *p* > 0.05). Waitlist participants experienced no change in knowledge over the study period (Tukey, *p* < 0.05). Implementation was also positively associated with control over enrichment implementation (Table 2).

Intent to provide rat tickling in the next year was significantly impacted by time (Figure 3b, Table 2). Compared to baseline, all treatment groups had higher intentions to tickle rats post-training and 2 months later (main effect of time, Tukey, *p* < 0.05). Intent to provide rat tickling in the next year was also positively associated with control over enrichment implementation and desire to provide more enrichment in general (Table 2).

At the third time point, all participants were asked to identify the picture which showed the most correct rat tickling technique. In this case, the picture of dorsal contact + pin would (Figure 2) be correct and all other answers would be incorrect. At this time point, significantly more participants in either trained group correctly identified the scientifically supported technique (96%), as compared to waitlist participants (73% correct; X^2^ (2, *n* = 86) = 9.7, *p* = 0.008; Figure 3c).

Implementation by individuals that participants currently supervised was significantly impacted by time (F_2,71_ = 30.6, *p* < 0.0001) but not treatment or their interaction (F_2,80_ = 1.6, *p* = 0.2; F_4,71_ = 1.5, *p* = 0.2). Supervisee implementation level was higher at the final 2-month follow-up survey compared to baseline or directly post-training (Tukey, *p* < 0.05).

#### 3.3.2. Knowledge, Self-Efficacy, and Familiarity

Factual knowledge of rat tickling was significantly associated by an interaction of treatment and time (Figure 4a, Table 3). Compared to baseline or the waitlist, both training groups had higher knowledge of rat tickling directly post-training and at the 2-month follow-up (Tukey, *p* < 0.05). Online-only training participants experienced a significant decrease in knowledge between post-training and the 2-month follow-up (although this remained higher than at baseline or compared to waitlist participants; Tukey, *p* < 0.05). Conversely, waitlist participants experienced no change in knowledge over the study period (Tukey, *p* < 0.05).

Rat tickling self-efficacy was significantly affected by the interaction of treatment and time (Figure 4b, Table 3). Compared to baseline, participants in both training groups increased in self-efficacy at post-training and the 2-month follow-up (Tukey, *p* < 0.05). Conversely, waitlist participants experienced no change in rat tickling self-efficacy over the study period (Tukey, *p* < 0.05). The only time point with significant differences between groups was directly post-training. At this time point, the online + hands-on training group had higher self-efficacy than the waitlist (Tukey, *p* < 0.05). Additionally, rat tickling self-efficacy was positively associated with baseline rat handling experience (Table 3).

Familiarity with rat tickling was significantly associated with an interaction of treatment and time (Figure 4c, Table 3). At baseline, groups were not significantly different in familiarity from each other. At post-training and the 2-month follow-up, compared to the waitlist, the online + hands-on group had higher familiarity than the waitlist (Tukey, *p* < 0.05). Compared to baseline, hands-on + online training participants reported an increase in familiarity post-training (Tukey, *p* < 0.05). Compared to baseline and the waitlist, both training groups reported an increase in their familiarity with rat tickling at the 2-month follow-up (Tukey, *p* < 0.05). Waitlist participants experienced no change in familiarity over the study period (Tukey, *p* < 0.05). Additionally, rat tickling familiarity was positively associated with baseline rat handling experience (Table 3).

#### 3.3.3. Beliefs and Human–Rat Interactions

Attitudes towards rat tickling were significantly associated with time point (Figure 5a, Table 4). Compared to baseline, all treatment groups had more positive attitudes directly post-training and at the 2-month follow-up (Tukey, *p* < 0.05). Additionally, more positive attitudes were seen in participants who more strongly agreed that they had a desire to provide more enrichment to their rats (Table 4).

Subjective norms towards rat tickling were significantly associated with time point (Figure 5b, Table 4). Compared to baseline, all treatment groups had more positive attitudes directly post-training and at the 2-month follow-up (Tukey, *p* < 0.05).

Perceived behavioral control to provide rat tickling was significantly associated with an interaction of treatment and time (Figure 5c, Table 4). Compared to baseline, hands-on + online training participants increased in perceived behavioral control post-training and the 2-month follow-up (Tukey, *p* < 0.05). Conversely, both online-only training and waitlist experienced no change in perceived behavioral control over the study period (Tukey, *p* < 0.05). Within each time point, no group was significantly different from the others (Tukey, *p* > 0.05). Additionally, perceived behavioral control was positively associated with desire to provide more enrichment, control over enrichment, and baseline rat handling experience (Table 4).

General human–rat interactions were not significantly associated with treatment, timepoint, or their interaction (*p* > 0.05). However, human–rat interactions were positively associated with control over enrichment and a desire to provide more enrichment (F_1,92_ = 6.8, *p* = 0.01; F_1,92_ = 5.9, *p* = 0.02).

### 3.4. Qualitative Data

Participant responses to open-ended questions about rat tickling were summarized into two central categories of benefits and barriers to rat tickling. These central categories were further split into themes and sub-themes, described below and summarized for trained individuals in Figure 6 and by group in Appendix A (which also includes additional representative quotes).

The majority of participants, regardless of training group, indicated that rat tickling was beneficial primarily for rat welfare (79% of all participants) and handling (63%). Benefits for personnel were also commonly listed (38%). Less commonly, participants also indicated benefits of rat tickling for research (9%), such as a better recovery time, more reliable physiologic reactions, and better-quality data. Within the category of rat welfare benefits, participants often specifically noted its benefits for reducing stress or anxiety (29%), providing enrichment (22%), and socialization (12%). Within the category of handling, some participants also mentioned the benefit of rat tickling to help develop a bond (7%). Several subcategories under personnel were mentioned. Most commonly participants indicated benefits to personnel well-being describing rat tickling as being fun, uplifting, and even reducing human stress (15%). Other potential benefits for personnel included increasing overall mood, empathy for animals, monitoring, and even approach to research/welfare overall. In fact, one participant indicated “…we’ve also noticed a measurable positive difference in the research personnel’s demeanor in the lab, their approaches to rat research, and deeper understanding of rat welfare needs.”

Participants identified several barriers preventing implementation of rat tickling, including time (44% of participants), personnel (43%), research (26%), and rat problems (14%). Within the category of time, some participants mentioned difficulties related to the quantity of rats, staffing limitations, time needed to train the technique to personnel, and consistency. Within the category of personnel, access to animals was commonly an issue (16%) as some participants simply were not directly involved with hands-on rat work and therefore personally may not be able to implement rat tickling. Additionally, participants mentioned a need to train individuals to do it properly (15%). Many individuals had concerns or difficulties with getting approval or buy-in for rat tickling. Within the category of research, most commonly, there was either concern regarding adding a new variable or research-related rat factors (e.g., head implants) that may prevent tickling rats for a certain period of time. Finally, in terms of rat factors, participants mentioned only having older rats or being concerned with individual differences or aggression. Barriers that were infrequently cited (2%) included having too few rats at an institution, small caging, or simply no barriers.

Beyond these central themes, participants in the trained groups made a few noteworthy comments about implementation and sharing the training module. That is, participants reported success in implementing rat tickling in their laboratories with a variety of different research paradigms, such as pharmacokinetic, diabetic, and tumor lesion rat models. Additionally, several participants mentioned that they were sharing the training module widely with colleagues and implementing it in their internal animal handling courses.

## 4. Discussion

To our knowledge, this is the first study to experimentally evaluate the efficacy of training laboratory animal personnel to improve important outcomes related to rat welfare. We compared training laboratory animal personnel about rat tickling via an interactive, highly visual, and online-only training module versus the same module supplemented with hands-on training, as compared to a waitlist control. We successfully sampled 97 participants at baseline and after training, with 86 of those completing a final survey two months later (i.e., 88% retention between post-training and the final survey).

Results indicated that training laboratory animal personnel with either online-only or online + hands-on modules was beneficial to important outcomes related to rat tickling implementation. At the end of the study as compared to baseline, trained personnel reported a higher frequency of implementation and significantly more personnel could correctly identify the scientifically supported method for rat tickling. Furthermore, at the end of the study compared to baseline, trained personnel had higher knowledge, self-efficacy, and familiarity with rat tickling.

The results from this study align with previous research and indicate that training personnel in rat tickling is beneficial for rat welfare and handling. For example, previous farm animal welfare recommendations from experts that targeted training can improve beliefs and implementation of positive handling techniques [14,23]. Additionally, previous research shows that more familiarity with rat tickling is strongly associated with implementation of the technique [3]. Further, more factual knowledge should help reduce perpetuation of misconceptions about rat tickling and ensure that it is applied in a scientifically supported manner. Finally, correct knowledge and implementation should help improve rat welfare and create a positive feedback loop when personnel observe these positive effects. Overall, training personnel in rat tickling has multiple benefits.

There were a few advantages to receiving the supplemental hands-on training over receiving only online training. First, only the online + hands-on training group showed increased perceived behavioral control compared to baseline. That is, personnel who went through both the online and hands-on training modules felt that tickling was easier to implement, more up to them, and overall felt more confident that they could provide rat tickling (as compared to baseline).

Second, at the end of the study, only the online + hands-on training group had significantly higher self-efficacy (i.e., confidence in tickling naïve/experience rats and doing all components of rat tickling) and familiarity with rat tickling compared to the waitlist. An explanation for these results could be that hands-on participants received immediate positive feedback from both the instructors and the bat detectors. Previous research shows that hands-on operant learning with immediate feedback is more effective than demonstration alone [24]. Additionally, they had an opportunity to tickle pre-trained rats rather than naïve rats, which is typically more challenging. Therefore, the hands-on participants can feel confident that their technique is correct and may also be more able to accurately assess themselves compared to the online-only training group. Our previous research in a sample of over 700 laboratory animal personnel indicates that perceived behavioral control shows a strong correlation with current implementation and intent to tickle rats [3].

Third, at the end of the study, the online + hands-on training group’s knowledge of rat tickling remained high while the online-only training group’s knowledge decreased slightly (though it remained higher than the waitlist and baseline). An explanation for these results could be that hands-on participants had their knowledge gained in the online course reinforced in the hands-on course, thereby providing more distributed learning and greater retention. This result may also support continued education in rat tickling for online-only training, in particular, although hands-on participants would likely also benefit from it.

Therefore, when feasible, we recommend also providing hands-on training in addition to the online training module because it improved perceived behavioral control and had greater benefits to self-efficacy and familiarity. However, if time and cost are prohibitive factors, then the online training module is still a reliable and effective training option that will contribute to increasing rat tickling implementation in the laboratory.

Although we did increase reported implementation of rat tickling in all groups and correct implementation in trained groups, at the end of the study trained personnel reported implementing rat tickling in less than 50% of the instances they interacted with rats. Although this may seem concerning, 50% of interactions may be a significant and effective increase, considering that rats positively respond to tickling after only 3 days of tickling for 15 s [12]. Furthermore, we did not see a significant difference between trained and waitlist participants at the end of the study. This could be due to self-reported measurement error (see Section 4.1), an insensitive scale, or even the Hawthorn effect as detailed in the final paragraph of this section. Future research on rat tickling implementation may need to specifically ask personnel to consider what percentage of rats they work with have been tickled for at least 3 days to capture the overall use of rat tickling within the laboratory and detect differences between treatment groups. However, it is also important to consider that rat tickling is not appropriate for all rats or models, particularly adult rats that have not been tickled previously.

Qualitative data overall supported rat tickling and training. The majority of participants indicated rat tickling was beneficial for rat welfare and handling (79% and 63%, respectively), which may reflect published research and corresponding presentations indicating these benefits [10]. Although less common, many participants (38%) also indicated that rat tickling is beneficial for personnel. This aligns with recent research showing that personnel who implement more diverse/frequent enrichment have lower reported burnout incidence [25]. Research was less frequently indicated as a benefit, which may be due to lack of knowledge in this area or because of a lack of principal investigators or other individuals who frequently design research studies participating in this study. It was also positive to note that trained participants noted success in using rat tickling with a wide variety of different research paradigms. Finally, trained participants noted that they plan to share the online course widely is a very positive indication of their support and the ability of the course to continue to spread post-study completion.

The qualitative data also indicate that some barriers to rat tickling still remain although no single barrier was mentioned by over half of participants. Time and personnel were most frequently cited by participants (44 and 43%, respectively), which is unsurprising. Even with recent research showing a 1000% decrease in the amount of time required to implement rat tickling [12], the technique does still require additional time to complete, especially when individuals are first being trained. Interestingly, some participants simply indicated that their particular roles did not often directly work hands-on with rats and therefore they could not implement rat tickling more often. However, these participants may still be important to support the work of others. Other participants indicated a need to train individuals to do it properly, which further supports the creation of the online training course. Finally, research and rat-related factors were indicated as barriers less frequently, but still a notable amount (26 and 14%, respectively). Certainly, some research paradigms that require rats to experience chronic variable stress may be contraindicated for rat tickling. Additionally, rats are known to have individual differences in their response to rat tickling and that it may be less effective with older rats [10]. Overall, these remaining barriers indicate that the time required to tickle rats and the personnel must still be supported to help increase rat tickling implementation.

At the end of this study, all participants, regardless of treatment, reported more positive attitudes, a higher intent to provide, and higher subjective norms related to rat tickling. This may be a result of the Hawthorne effect, in which participants alter their behavior simply due to their awareness of being observed. In this study, simply by being asked about rat tickling, waitlist participants may have sought out additional information about rat tickling during the study period that may have changed their opinions or encouraged them to start attempting implementation. For example, our team has published an online downloadable handout, a video protocol, and a general information video that are all freely available online (ag.purdue.edu/ansc/gaskill/resources/) [10,12,26]. The Hawthorne effect could have also played a part in why our study did not detect differences in implementation between waitlist and trained participants at the end of the study. Furthermore, as participants were asked about their intent to tickle rats over the next year, we may not have seen significant differences since most waitlist participants appeared eager to take the training module once the study period was completed. Regardless, the positive results seen in all treatment groups seem to be a positive indicator for establishing rat tickling as a more common intervention in the laboratory.

### 4.1. Limitations

This study is not without limitations. First of all, since this study only involved self-report data there is the potential for measurement errors and subjective biases to occur. For example, participants may have inaccurately rated their own implementation, familiarity, and other important outcomes. From experience, we know that some people think that simply scratching a rat on the back of the neck is rat tickling. Therefore, participants, especially at baseline or those assigned to the waitlist groups, may have been unable to accurately rate themselves. Furthermore, trained participants may have become more conservative in their ratings as they became more knowledgeable while waitlist participants over-reported due to the Dunning–Kruger effect, where less trained individuals tend to over-report their competence. Thus, if trained participants became more conservative and waitlist participants over-report, then our results may be even stronger than they appear, although we cannot say for sure. This potential measurement error is also a possible explanation for why implementation of rat tickling was not significantly different between groups at the end of this study. However, despite these limitations, our study still provides support for targeted training being beneficial to important outcomes related to rat tickling.

Second, although there is strong scientific evidence for the benefits of rat tickling [10], animal-based measures of welfare were not assessed in this study. Therefore, we cannot definitely claim that rat welfare was improved simply because of increased implementation. After all, even trained participants may not correctly apply the technique or self-report their application accurately. Furthermore, even if rat tickling is applied correctly and rated accurately, its application may not ubiquitously increase positive affect and welfare; rats show individual differences in response to tickling and are less responsive when stressed or older [10]. However, participants were educated about these factors, taught to evaluate rat responses to tickling, and discontinue if negative responses are continued to be observed. Therefore, if participants followed trained procedures correctly, it is likely, albeit not proven, that rat welfare was improved. Overall, this study still provides valuable insight in the efficacy of training laboratory animal personnel.

Finally, an additional limitation is that this study was not fully randomized at baseline. If participants were able to participate in a hands-on training session, then they were assigned to that group. Therefore, the hands-on training group may be skewed towards individuals with more flexibility in their schedules or baseline support from their supervisors to attend such a hands-on session. This could indicate that these participants have more potential for an increase in implementation or perceived behavioral control of rat tickling over time. However, as there was a major geographic restriction for hands-on workshop participants, there were likely individuals from other areas that would have attended the workshop if it was possible, therefore minimizing this possible difference. Furthermore, no outcome variables were significantly different at baseline. The only control variable that was significantly different at baseline was the level of stress/pain of rats, which has not been shown to influence outcome measures in this study or our previous work [3]. Therefore, despite these limitations, there is still good support for the positive outcomes of rat tickling training.

## 5. Conclusions

In conclusion, targeted training of laboratory animal personnel in rat tickling in either online-only or online + hands-on formats is effective in improving implementation, knowledge, self-efficacy, and familiarity with rat tickling. Hands-on training was more beneficial as it also improved perceived control beliefs and had greater benefits for self-efficacy and familiarity. Therefore, if possible, online + hands-on training will provide the most benefits. However, if hands-on training is not possible, then online-only training is still beneficial.

It is also important to note that our online training module was developed after teaching seven hands-on workshops to diverse audiences over three years. Furthermore, the online training module included multiple interactive elements, extensive video and pictorial examples, and addressed common difficulties when implementing the technique. A less interactive or detailed online training module may not have the same effects.

Overall, this study demonstrates that curated interactive training courses in rat tickling (i.e., online + hands-on training, if possible, but otherwise online-only training) are an effective means for promoting implementation of animal welfare-enhancing techniques.

## Figures and Tables

**Figure 1 animals-10-01435-f001:**
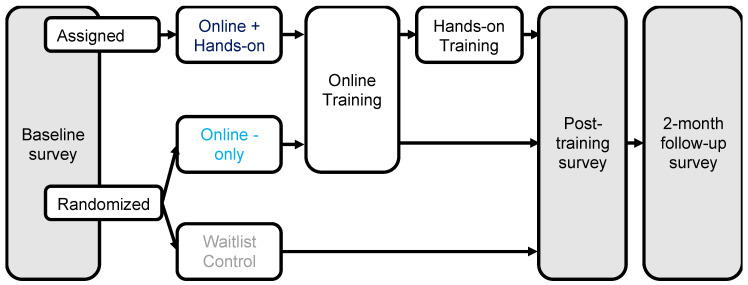
Timeline of procedures and assessment for laboratory animal personnel. All participants were given a baseline survey before group assignment. Then, they were either assigned or randomized to groups and then received training. Immediately after training (or a similar time point for waitlist), participants received a post-training survey. Two months later, they received a final follow-up survey.

**Figure 2 animals-10-01435-f002:**
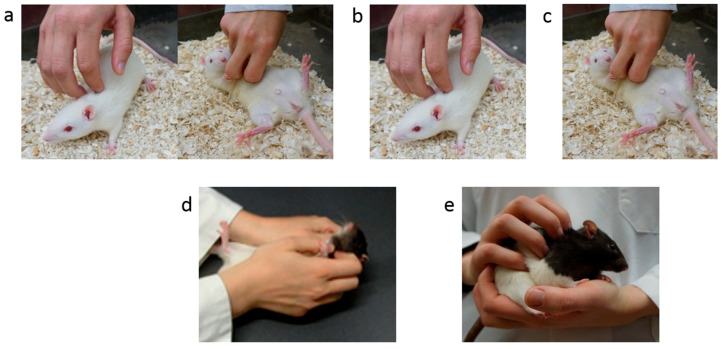
Pictorial tickling procedures. (**a**): Dorsal contact and pin (standard, validated rat tickling procedure), (**b**): Dorsal contact only or stroking in the cage, (**c**): Pin only, (**d**): Two-handed pin only, (**e**): Stroking in the hand.

**Figure 3 animals-10-01435-f003:**
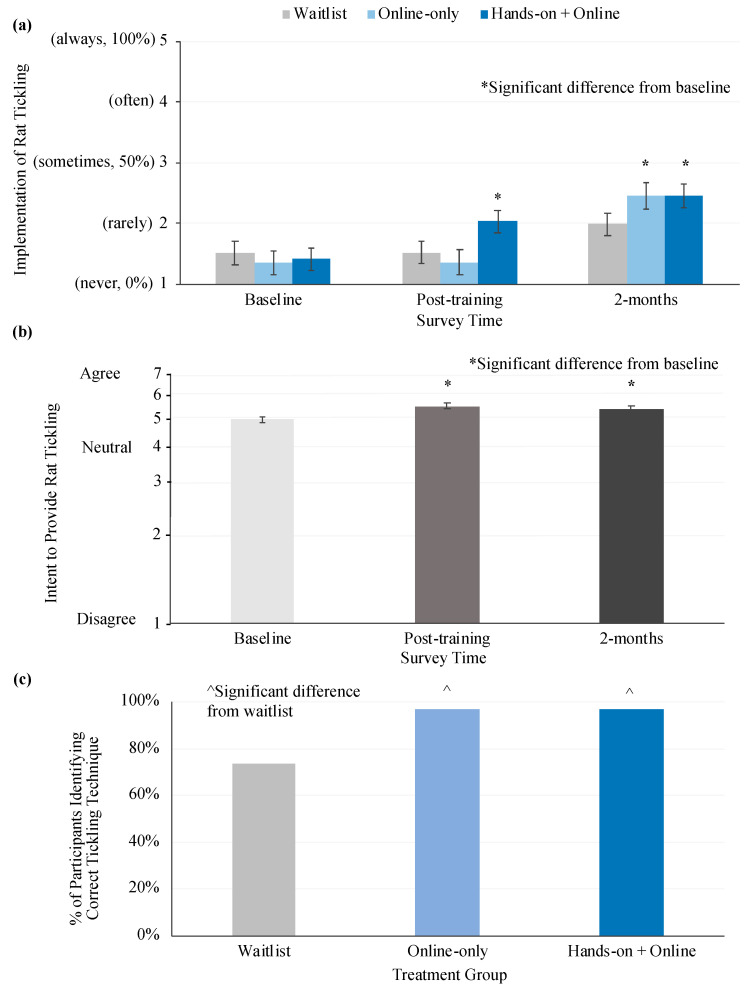
Implementation and intent to provide rat tickling/correct identification of technique. (**a**) and (**b**) This figure shows the highest order significant associations from general linear mixed models that impacted participants current implementation and intent to provide rat tickling. Implementation and intent were measured via a self-report survey. Models were run controlling for potential confounding variables. Both scales display only the range of possible responses. Data is presented as least squares mean ± standard error of the mean. The scale of intent is back transformed from log10. (a) * indicates a significant difference from baseline within treatment group. (b) * indicates a significant difference from baseline. (**c**) The bottom graph shows the percentage of participants that selected a dorsal contact + pin picture when asked to identify which picture indicates the correct rat tickling technique. ^ indicates a significant difference from waitlist.

**Figure 4 animals-10-01435-f004:**
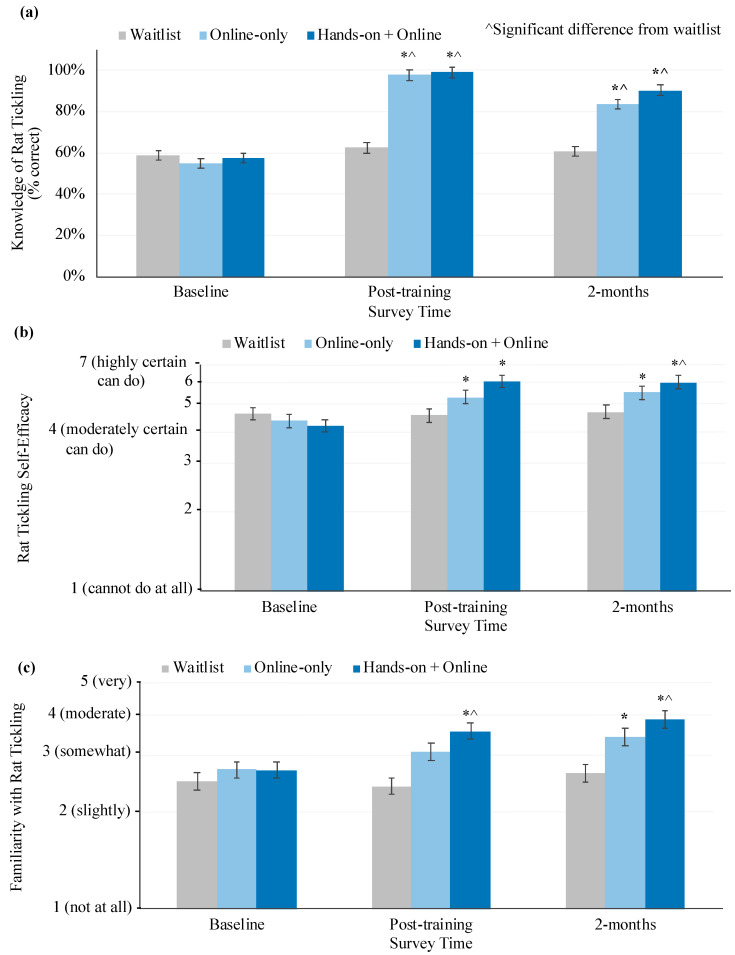
Knowledge, self-efficacy, and familiarity with rat tickling. This figure shows the highest order significant associations from general linear mixed models of (**a**) knowledge, (**b**) self-efficacy, and (**c**) familiarity with rat tickling. Variables were measured via a self-report survey. Models were run controlling for demographic, work, and potential confounding variables. The scales of (b) self-efficacy and (c) familiarity are back transformed from log10. Scales display only the range of possible responses. Data is presented as least squares mean ± standard error of the mean. * indicates a significant difference from baseline within the treatment group. ^ indicates a significant difference from the waitlist within the time point.

**Figure 5 animals-10-01435-f005:**
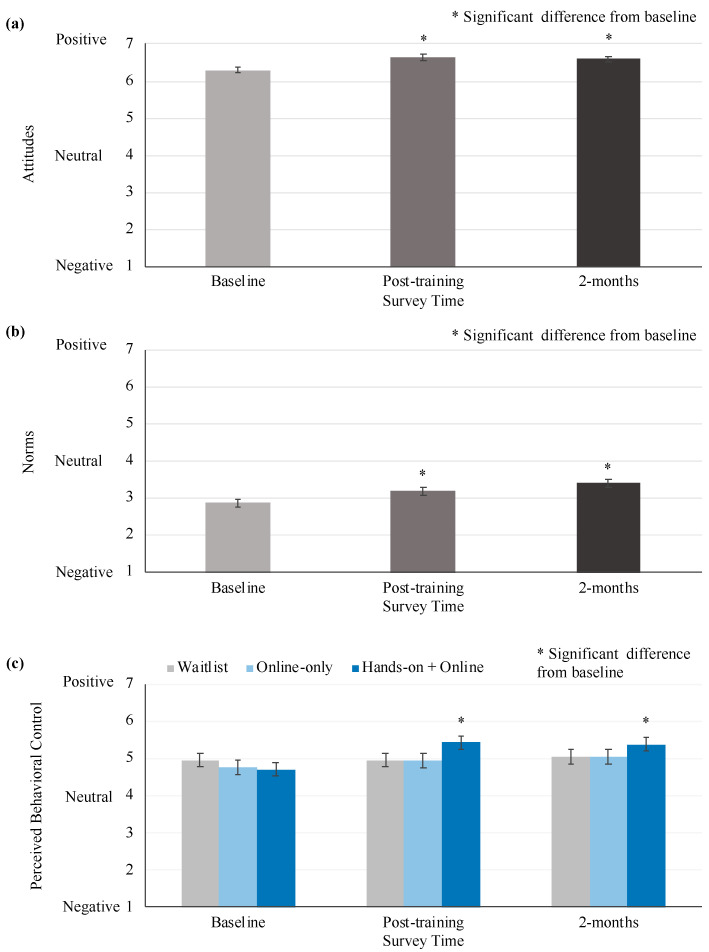
Beliefs about rat tickling. This figure shows the highest order significant associations from general linear mixed models of (**a**) attitudes, (**b**) subjective norms, and (**c**) perceived behavioral control with rat tickling. Variables were measured via a self-report survey. Models were run controlling for potential confounding variables. Scales display only the range of possible responses. Data is presented as least squares mean ± standard error of the mean. (a & b) * Indicates a significant difference from baseline. (c) * Indicates a significant difference from baseline within the treatment group.

**Figure 6 animals-10-01435-f006:**
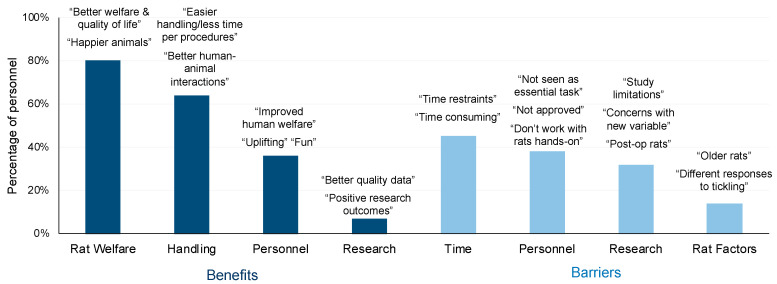
Trained personnel—benefits and barriers to rat tickling. The most common themes related to benefits (advantages) and barriers (factors making it difficult) to tickle rats reported by 86 laboratory animal personnel in a 2-month follow-up survey. Graphic includes representative quotes. Sub themes and additional representative quotes are presented in Appendix A.

**Table 1 animals-10-01435-t001:** Baseline characteristics of laboratory animal personnel across treatment groups (*n* = 97).

Continuous Data (Range)	Group (M ± S.D.)			Difference
	Online + Hands-on	Online-Only	Waitlist	
Rat Stress/Pain (1–4)	2.4 ± 0.1	2.1 ± 0.1	1.8 ± 0.1	F_2,88_ = 5.9, *p* = 0.004
Enrichment Control (1–5)	3.2 ± 0.2	3.2 ± 0.2	3.1 ± 0.2	F_2,93_ = 0.2, *p* = 0.8
Enrichment Desire (1–7)	5.4 ± 0.2	5.7 ± 0.3	4.9 ± 0.3	F_2,93_ = 2.5, *p* = 0.09
Rat Handling Experience (1–7)	5.7 ± 0.2	5.9 ± 0.2	6.0 ± 0.2	F_2,93_ = 0.8, *p* = 0.4

As treatment groups were semi-randomized, they were tested at baseline for the possibility of significant differences in baseline characteristics via analysis of variance (F-tests). M = mean, S.D. = standard deviation.

**Table 2 animals-10-01435-t002:** Implementation and intent to provide rat tickling. The associations from general linear mixed models of self-reported data from laboratory animal personnel. Participants were asked about their current implementation and intent to provide rat tickling to their laboratory rats.

Independent Variable	Implementation	Intent
Timepoint	**F_2,154_ = 31.0, *p* < 0.0001**	**F_2,178_ = 13.8, *p* < 0.0001**
Treatment	F_2,163_ = 0.16, *p* = 0.9	F_2,165_ = 1.3, *p* = 0.26
Timepoint × Treatment	**F_4,154_ = 3.5, *p* = 0.009**	F_4,178_ = 1.2, *p* = 0.3
Control over Enrichment	F_1,90_ = 5.5, *p* = 0.02	**(+) F_1,92_ = 6.4, *p* = 0.01**
Enrichment Desire		**(+) F_1,92_ = 6.9, *p* = 0.01**

Blank cells indicate that the covariate was not included in the final model. Timepoint: Pre-training, post-training, and 2-month follow-up post-training. Treatment: Online-only training, online + hands-on training, or waitlist control. Bold indicates a significant effect with *p* < 0.05. (+) indicates a positive association.

**Table 3 animals-10-01435-t003:** Knowledge, self-efficacy, and familiarity with rat tickling. The associations from general linear mixed models of self-reported data from laboratory animal personnel. Participants were asked about their factual knowledge, self-efficacy, and familiarity with rat tickling.

Independent Variables	Knowledge	Self-Efficacy	Familiarity
Timepoint	**F_2,179_ = 161.7, *p* < 0.0001**	**F_2,177_ = 24.5, *p* < 0.0001**	**F_2,178_ = 18.8, *p* < 0.0001**
Treatment	F_2,234_ = 0.7, *p* = 0.5	F_2,187_ = 0.8, *p* = 0.5	F_2,186_ = 0.6, *p* = 0.5
Timepoint × Treatment	**F_4,179_ = 32.0, *p* < 0.0001**	**F_4,177_ = 7.4, *p* < 0.0001**	**F_4,178_ = 4.5, *p* = 0.002**
Rat Handling Experience		**F_1,90_ = 16.0, *p* = 0.0001**	**(+) F_1,91_ = 9.8, *p* = 0.002**

Blank cells indicate that the covariate was not included in the final model as *p* > 0.05. Timepoint: Pre-training, post-training, and 2-month follow-up post-training. Treatment: Online-only training, online + hands-on training, or waitlist control. Bold indicates a significant effect with *p* < 0.05. (+) indicates a positive association.

**Table 4 animals-10-01435-t004:** Beliefs about rat tickling. The associations from general linear mixed models of self-reported data from laboratory animal personnel. Participants were asked about their attitudes, subjective norms, and perceived behavioral control with rat tickling.

Independent Variables	Attitudes	Subjective Norms	Perceived Behavioral Control
Timepoint	**F_2,178_ = 13.5, *p* < 0.0001**	**F_2,179_ = 14.0, *p* < 0.0001**	**F_2,178_ = 8.3, *p* = 0.0003**
Treatment	F_2,176_ = 1.2, *p* = 0.3	F_2,177_ = 1.0, *p* = 0.2	F_2,144_ = 0.5, *p* = 0.6
Timepoint × Treatment	F_4,178_ = 2.4, *p* = 0.05	F_4,179_ = 2.1, *p* = 0.07	**F_4,178_ = 3.5, *p* = 0.009**
Enrichment Desire	**(+) F_1,92_ = 15.3, *p* = 0.0002**		**(+) F_1,91_ = 4.5, *p* = 0.04**
Control over Enrichment			**(+) F_1,91_ = 7.5, *p* = 0.008**
Rat Handling Experience			**(+) F_1,90_ = 5.1, *p* = 0.03**

Blank cells indicate that the covariate was not included in the final model. Timepoint: Pre-training, post-training, and 2-month follow-up post-training. Treatment: Online-only training, online + hands-on training, or waitlist control. Bold indicates a significant effect with *p* < 0.05. (+) indicates a positive association.

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
