# Peer review of "Changing Human Behavior to Improve Animal Welfare: A Longitudinal Investigation of Training Laboratory Animal Personnel about Heterospecific Play or “Rat Tickling”"

_animals, 2020, doi:10.3390/ani10081435_

Round 1

Reviewer 1 Report

This is a very nicely written article on a topic that is timely and infrequently discussed in the literature.  The methodology is sound, with an appropriate control group.  The writing style is clear and easy to understand.  I have no comments to make other than great job!

Author Response

We would like to thank Reviewer 1 for the positive comments. We are so glad you enjoyed the paper.

Reviewer 2 Report

Overall, this is a well-written paper, and for its objectives a decently executed study. This said, it is important to contend with key presuppositions and generalizations underlying the paper itself. Key presuppositions are that rat tickling is a) universally positive that b)can and should be generally operationalized.

The evidence to suggest positive affect is rat vocalization in the positive range (laughing) and positive anticipation of tickling. The rats that do laugh have also shown positive affective bias. However, this is not true for each rat. Rat tickling as developed by Panksepp, was used to simulate conspecific play. Inherent in this, is the that rats are relational, and play is a manifestation of choice and agency within that relationship.

As Panksepp himself stated, rats who are stressed won’t laugh. Any effective training with the goal of positive welfare must account for personality differences among the rats, which would include perception of stress/stressor and the reception of tickling itself, as well as the development of a relationship between the rat and the handler. Description of the handling technique itself needs to be contextualized with regards to these issues (for example, there is evidence to suggest tickling should be done on the belly, but prescribing this in isolation without addressing individuality/relationship is problematic).

The paper does not include any welfare assessments post training treatment, presumably for the presupposition that rat tickling is a positive handling technique.

At the outset and further in the Discussion section, the present paper needs to address these concerns of the presuppositions, which speaks to the limitations of the scientific findings regarding the technique itself and potential limitations of any training methods.

Thus, the training methods in substance and across modality need to be discussed in light of such limitations. Did the training methods used in the study make any attempts to address relationality and individual variation of rats? Is implementation in and of itself proof of positive effect or proper implementation?

Line 88: What would be considered “proper training”?

Line 96: Is it considerably? Evidence for the presumed variation?

Fig 1 - Some wonkiness in the arrows.

Lines 186-189: Can the authors speak to the changes made to the training

Line 213: If a rat is presenting as difficult to handle does this suggest it is not receptive to tickling for reasons to be determined?

What is the advice given when rats are not physically receptive, when the laughing is not expressed?

Lines 218-222: Can the authors expand on the specifics of the use of non-validated instruments, etc.

Line 241: Rephrase for clarity. “Rat tickling current and planned use were assessed by a few key questions”

Tables 1, 4, 5 need some formatting modification.

Author Response

Overall, this is a well-written paper, and for its objectives a decently executed study. This said, it is important to contend with key presuppositions and generalizations underlying the paper itself. Key presuppositions are that rat tickling is a) universally positive that b)can and should be generally operationalized.

The evidence to suggest positive affect is rat vocalization in the positive range (laughing) and positive anticipation of tickling. The rats that do laugh have also shown positive affective bias. However, this is not true for each rat. Rat tickling as developed by Panksepp, was used to simulate conspecific play. Inherent in this, is the that rats are relational, and play is a manifestation of choice and agency within that relationship.

As Panksepp himself stated, rats who are stressed won’t laugh. Any effective training with the goal of positive welfare must account for personality differences among the rats, which would include perception of stress/stressor and the reception of tickling itself, as well as the development of a relationship between the rat and the handler. Description of the handling technique itself needs to be contextualized with regards to these issues (for example, there is evidence to suggest tickling should be done on the belly, but prescribing this in isolation without addressing individuality/relationship is problematic).

The paper does not include any welfare assessments post training treatment, presumably for the presupposition that rat tickling is a positive handling technique.

At the outset and further in the Discussion section, the present paper needs to address these concerns of the presuppositions, which speaks to the limitations of the scientific findings regarding the technique itself and potential limitations of any training methods. Thus, the training methods in substance and across modality need to be discussed in light of such limitations.

Did the training methods used in the study make any attempts to address relationality and individual variation of rats?

  • We would like to thank reviewer 2 for their feedback. We would like to encourage them to review our certificate course as we believe it addresses concerns about designing effective training that accounts for individual differences in the rats as well as potential current stress levels. See bit.ly/RatTicklingCertificate and then navigate to the open course. Originally, we did not include many details of the course in the paper in order to keep it focused and brief. However, we have now added more detail and context to the paper to address your concerns.
  • We do not include any welfare assessment post-training treatment as the purpose of the study was not to assess the effectiveness of rat tickling, but rather the effectiveness of training modalities on human related outcomes.
  • We have added more information throughout the paper addressing concerns about individual differences and stress levels. We encouraged participants to implement rat tickling as their initial interaction with their rats before any stressful procedures are performed. Furthermore, we emphasize that there are individual differences and that some rats may not respond positively to the technique.
  • The training methods addressed the individual variation of rats and potential consequences of stressful environments on rat tickling. We have added further information on this to section 2.2.1 Online Training
    • Course topics included the rationale behind rat tickling, detailed pictorial/video instruction on the hands-on technique (including videos of both what to do and what not to do), guidance on implementation, how to assess a rat’s response to the technique (including videos of positive and negative rat reactions to tickling) and respond appropriately, and a series of frequently asked questions. Assessing a rat’s response to the technique is particularly important since rats show a wide degree of individual differences in response to rat tickling [LaFollette 2017]. Based on previous literature, we recommend tickling juvenile rats for at least 3 days [LaFollette 2019] before assessing unless overt aggression or extreme distress is seen. This allows rats to understand the technique and gain any possible habituation or handling benefits. Additionally, we cautioned participants on implementing tickling in older rats, highly stressed rats, and breeder rats.
  • However, we do not know of any literature to support relationality and rat tickling. In our own studies and most previous literature, rat tickling is the singular intervention and not paired with other interventions to increase relationality. Therefore, we did not add any information on this. However, if the reviewer can provide a citation detailing this we would be happy to add it to the discussion.
  • We also do acknowledge that rat tickling is not always the appropriate intervention in the discussion section Line 636-638: “However, it is also important to consider that rat tickling is not appropriate for all rats or models, particularly adult rats that have not been tickled previously.”

Is implementation in and of itself proof of positive effect or proper implementation?

  • We had previously acknowledged in the limitations section that participants may be unable to accurately rate their own implementation with rat tickling and they could be doing in incorrectly, particularly in untrained groups. However, we now have added sentences to the limitations section that acknowledge that even after training participants could still be improperly applying rat tickling and that implementation itself does not necessarily equal proof of positive affect.
    • “Alternatively, it is possible that even trained groups may inaccurately rate their own implementation and could still be improperly applying rat tickling. And even if rat tickling is applied using correct technique and rated accurately, simple application may not lead to increased positive affect of rats because of individual differences or if applied to older or stressed rats [LaFollette 2017].”

Line 88: What would be considered “proper training”?

  • Line 88: We have removed the word “proper” as it is more accurate to say that participants simply said that any training itself is not available.

Line 96: Is it considerably? Evidence for the presumed variation?

  • In large-scale animal agriculture, a high degree of standardization is used to increase efficiency. Most animal farms contain only a single species, in standardized housing, sometimes even only a single age or sex, typically used for a single purpose (e.g., meat production versus egg production). This is in great contrast to a research laboratory which typically contains several species used for many diverse studies. Research animals have much more heterogenous handling and housing simply by design. We have added the wording
    • “…large-scale animal agriculture, which typically houses a single species for a singular purpose with homogenous housing such as egg-laying hens in an aviary system”

Fig 1 - Some wonkiness in the arrows.

  • Fig 1 – We have fixed the arrows.

Lines 186-189: Can the authors speak to the changes made to the training

  • We made quite a few changes to our training methods over the years. Here are a few examples. In the hands-on course, we added moving our fingers on the back of participants’ hands to guide the pressure needed during tickling. In the online course/powerpoint, we added updated and more detailed pictures and videos of the flipping method. We also clarified what we feel is the easiest to learn/most effective method of the flip. We have added a singular example of these changes within the text:
    • For example, we have repeatedly found that the transition from dorsal contact to pin is the most difficult part of rat tickling for students. Therefore, we added extensive pictorial, video, and hands-on instruction on this technique and clarified a very specific recommended method for this transition that we have found to be the most effective for most students.”

Line 213: If a rat is presenting as difficult to handle does this suggest it is not receptive to tickling for reasons to be determined?

  • Based on prior research, some rats will not make positive vocalizations during their first rat tickling session, but subsequently make numerous positive vocalizations after 3 days of tickling. However, on that first day, rats can even display some negative behavioral reactions to tickling. To us, it seems that at first rats may not comprehend that the interaction is playful, but with repeated exposure they seem to understand this. Therefore, naïve rats are typically more difficult to handle, although even on day 1 with some rats are taking to the intervention immediately. We give some information on this in the course. Within the paper text we have added more information to this section:
    • On the first day of tickling it is not unusual for rats to produce fewer 50-kHz vocalizations and be somewhat resistant as it is an entirely new interaction with human. However, after 3 tickling sessions most juvenile rats habituate and show behavioral signs of positive affect [LaFollette 2019].”

What is the advice given when rats are not physically receptive, when the laughing is not expressed?

  • We recommend, based on prior research and our own experience, that participants typically implement rat tickling for 3 days before necessarily evaluating rat behavior. However, we also recommend that participants use their best judgment and consider the rats as individuals and their age/stress status. This is all covered in the course.
  • We have added some of this information to the online course section of the article detailing some of these concerns and how they were addressed.
    • Course topics included the rationale behind rat tickling, detailed pictorial/video instruction on the hands-on technique (including videos of both what to do and what not to do), guidance on implementation, how to assess a rat’s response to the technique (including videos of positive and negative rat reactions to tickling) and respond appropriately, and a series of frequently asked questions. Assessing a rat’s response to the technique is particularly important since rats show a wide degree of individual differences in response to rat tickling [LaFollette 2017]. Based on previous literature, we recommend tickling juvenile rats for at least 3 days [LaFollette 2019] before assessing unless overt aggression or extreme distress is seen. This allows rats to understand the technique and gain any possible habituation or handling benefits. Additionally, we cautioned participants against tickling in older rats, highly stressed rats, and breeder rats.”

Lines 218-222: Can the authors expand on the specifics of the use of non-validated instruments, etc.

  • We have listed the model name and number and the frequency they were set to. Bat detectors have previously been used in research to quantify rat ultrasonic vocalizations (see Panksepp and Burgdorf, 2000). Their main disadvantages are that they can only capture a very narrow frequency range (positive vocalizations – trills - can occur between 35-75 kHz), they do not visualize the calls, and they do not record for playback. We have added,
    • These devices transduce ultrasonic vocalizations at a very specific level into a range audible to humans in real time. Similar devices were used in the original rat tickling experiments [Panksepp and Burgdorf 2000]. Although these devices do not capture the full range of possible vocalizations (20-75kHz) they are very simple to use and can be obtained at a significantly lower price than other ultrasonic microphones. Therefore, we recommend to participants to purchase such detectors for their laboratories.”

Line 241: Rephrase for clarity. “Rat tickling current and planned use were assessed by a few key questions”

  • Rephrased to “A few key questions were used to assess participants’ current and future implementation of rat tickling.”

Tables 1, 4, 5 need some formatting modification.

  • We have fixed indentation on Table 4 and Table 5. Table 1 has been moved to supplemental materials based on Reviewer 3’s suggestion.

Reviewer 3 Report

Summary 

The aim of this paper is a better understanding of animal welfare education (theoretical and practical) and its effects in different profiles in laboratory animals facilities with the specific topic of rat tickling.  

The main contributions describe a significant effect of online training and online + hands on training, in the use of rat ticking and knowledge about this technique.  

Broad comments  

The study is based in online surveys in three different times (baseline; post survey and 2 months later) 

The results support the conclusions, but I found that some aspects could clarify the message of the paper.  

  1.  For clarity I suggest that some of the information presented in the main text, could be placed as supplementary data to reduce the length of the text. For example: 
  • table 1 
  • Data from figure 3b and c seem difficult to interpret including all treatments and/ or times.  

2.  I missed in the results and specially in the discussion and the limits of the study some comments about why implementation of rat tickling was not significantly different two months  after the study between groups  (even if it was within the same group between periods).  

3. In the figure 1 there is no board-certified veterinarian in the waitlist. Is this due to the randomized procedure or assignments? Please specify this case, as graduation could be an important bias in the groups 

Specific comments 

1  L211 Please specify if any specific strain was used during all training sessions (and specify, with its complete genotypic nomenclature or different strains were used)

2  L213 Please give the complete reference of the device (bat detector) 

3  L299 The word survey is repeated  

4  L325 participant (treatment) instead of participant(treatment) 

5  Table 1 in the column group-online only, and the information about the race includes only 33 per cent of responses, is it a mistake? If not, why the rest are not completed as “prefer not to answer”? 

6  L359 Please give the year of publication and the full reference at the end of the paper with the other bibliographic references.  

7  L378 Please include in the caption of the figure the intervals used for these parameters (minimum-maximum), as it is difficult to interpret this information alone without these values.  

8  L381 Figures 3a and 3b the data is showed as the mean-median? With the SEM or SD? Please include this information in the captions  

9  L426 this decrease in knowledge could be used as a support for continued education in animal welfare in the discussion 

10  L470 I suggest erasing background lines or use them in all figures for homogeneity in the presentation of the results.  

Author Response

Summary 

The aim of this paper is a better understanding of animal welfare education (theoretical and practical) and its effects in different profiles in laboratory animals facilities with the specific topic of rat tickling.  

The main contributions describe a significant effect of online training and online + hands on training, in the use of rat ticking and knowledge about this technique.  

Broad comments  

The study is based in online surveys in three different times (baseline; post survey and 2 months later).  

The results support the conclusions, but I found that some aspects could clarify the message of the paper.  

  • We would like to thank reviewer 3 for their feedback and appreciate their suggestions to clarify messaging to enhance the manuscript.
  1. For clarity I suggest that some of the information presented in the main text, could be placed as supplementary data to reduce the length of the text. For example: table 1 & Data from figure 3b and c seem difficult to interpret including all treatments and/ or times.  
  • Table 1 has been moved to supplementary data. However, we feel it is important to include graphical representation of significant results for Figure 3b and 3c. Also, as only the main effects were significant (not the interactions) we present only those main effects.
  1. I missed in the results and specially in the discussion and the limits of the study some comments about why implementation of rat tickling was not significantly different two months after the study between groups (even if it was within the same group between periods). 
  • Within the results section about self-reported implementation we state,
    • Within each time point, no group was significantly different from the others (Tukey, p’s > 0.05).”
  • We believe there are a couple possible reasons that implementation of rat tickling was not significantly different at the end of the study. We have added this information to the discussion lines 629-632 –
    • Furthermore, we did not see a significant difference between trained and waitlist participants at the end of the study. This could be due to self-reported measurement error (see Limitations section 4.1), an insensitive scale, or even the Hawthorn effect as detailed in the final paragraph of this section.”
  • First in section 4.1. Limitations, we address possible self-reported measurement error. It is possible that waitlist participants may be likely to over-report their own implementation level throughout the study while trained participants may become more conservative in their ratings over time due to increased accuracy of knowledge. We have added this sentence for clarity
    • This potential measurement error is a possible explanation for why implementation of rat tickling was not significantly different between groups at the end of this study.”
  • Second, there is the possibility that there is a scale measurement error as detailed in lines 632-635. That is, our scale, may not have been sensitive enough with small increments to capture differences. We have added,
    • “Future research on rat tickling implementation may need to specifically ask personnel to consider what percentage of rats they work with have been tickled for at least 3 days to capture the overall use of rat tickling within the laboratory and detect differences between treatment groups.”
  • Third, there is a potential Hawthorne effect in which participants alter their behavior simply by being aware that they are being observed. There is quite a bit of open access resources on rat tickling that the waitlist could have garnered. Once they signed up for a “rat tickling” study and knew they would receive training at the end of it, they may have been triggered to search and learn more about the topic in preparation or due to enthusiasm to participate. This effect is detailed as an explanation for why all participants experienced increases in some outcomes.
    • We have added this sentence to the larger paragraph, “The Hawthorne effect could have also played a part in why our study did not detect differences in implementation between waitlist and trained participants at the end of the study.”
    • Here is the larger paragraph for reference, “At the end of this study, all participants regardless of treatment, reported more positive attitudes, a higher intent to provide, and higher subjective norms related to rat tickling. This may be a result of the Hawthorne effect, in which participants alter their behavior simply due to their awareness of being observed. In this study, simply by being asked about rat tickling, waitlist participants may have sought out additional information about rat tickling during the study period that may have changed their opinions or encouraged them to start attempting implementation. For example, our team has published an online downloadable handout, a video protocol, and a general information video that are all freely available online (ag.purdue.edu/ansc/gaskill/resources/) [10,12,25]. The Hawthorne effect could have also played a part in why our study did not detect differences in implementation between waitlist and trained participants at the end of the study. Furthermore, as participants were asked about their intent to tickle rats over the next year, we may not have seen significant differences since most waitlist participants appeared eager to take the training module once the study period was completed. Regardless, the positive results seen in all treatment groups seem to be a positive indicator for establishing rat tickling as a more common intervention in the laboratory.”
  1. In the figure 1 there is no board-certified veterinarian in the waitlist. Is this due to the randomized procedure or assignments? Please specify this case, as graduation could be an important bias in the groups.  
  • That there is no board-certified veterinarian in the waitlist occurred simply due to randomization. We used a true randomization protocol between waitlist and online-only rather than trying to balance groups based on different characteristics. All of those individuals with a veterinary degree did indeed graduate. Board certified veterinarians completed an additional board certification such as the American College of Laboratory Animal Medicine or American College of Animal Welfare. We have added this detail to what is now Table S2. Furthermore, there were no significant differences between groups in overall certification (p = 0.4) or education level (p = 0.9).
  • Furthermore, since at baseline there were no differences in outcome measures between groups and it was a longitudinal study, we do not believe this difference to bias the groups.

Specific comments 

1  L211 Please specify if any specific strain was used during all training sessions (and specify, with its complete genotypic nomenclature or different strains were used)

  • We have added the following text The rats used in training session varied due to host institutional needs, participant sign-ups, and to apply the 3Rs principle of reduction to minimize additional rats that would not find subsequent use. At the two training locations that could find subsequent use for new rats, 20 juvenile 6-week-old Long-Evans rats (Crl:LE) were ordered for the workshop. At the other locations, we used available surplus rats from studies or breeding colonies (i.e., ten 8-week-old Crl:CD rats and at the final location four 8-week-old and four 12-week-old Crl:CD).”

2  L213 Please give the complete reference of the device (bat detector) 

  • We have added the complete reference (Bat 4 Bat Detector, Magenta Electronics Ltd., Staffordshire, UK)”

3  L299 The word survey is repeated  

  • The excess word is deleted.

4  L325 participant (treatment) instead of participant(treatment) 

  • A space is added between participants (treatment).

5  Table 1 in the column group-online only, and the information about the race includes only 33 per cent of responses, is it a mistake? If not, why the rest are not completed as “prefer not to answer”? 

  • This is a mistake. We fixed it with correct percentages and double checked the rest of the table as well.

6  L359 Please give the year of publication and the full reference at the end of the paper with the other bibliographic references.  

  • We have added the citation within the methods section when the scale is first mentioned.
    • Full reference = Office of Animal Care and Use. National Institutes of Health. (2017). Guidelines for Preparing USDA Annual Reports and Assigning USDA Pain & Distress Categories, https://oacu.oir.nih.gov/sites/default/files/uploads/arac-guidelines/usda.pdf

7  L378 Please include in the caption of the figure the intervals used for these parameters (minimum-maximum), as it is difficult to interpret this information alone without these values.  

  • We have added the minimum and maximum values for each baseline characteristic within the table itself to aid in interpretation.

8  L381 Figures 3a and 3b the data is showed as the mean-median? With the SEM or SD? Please include this information in the captions  

  • We have added Data is presented as least squares mean ± standard error of the mean.” To the figure legends

9  L426 this decrease in knowledge could be used as a support for continued education in animal welfare in the discussion 

We have added more information on this result in the discussion including the reviewers suggestion Third, at the end of the study, the online + hands-on training group’s knowledge of rat tickling remained high while the online-only training group’s knowledge decreased slightly (though it remained higher than the waitlist and baseline). An explanation for these results could be for hands-on participants the knowledge gained in the online course was reinforced in the hands-on course thereby providing more distributed knowledge and greater retention. This result may also support continued education in rat tickling for online-only training, in particular, although hands-on participants would likely also benefit from it.”

10  L470 I suggest erasing background lines or use them in all figures for homogeneity in the presentation of the results. 

  • We added background lines to Figure 6 although we made them significantly lighter to accommodate the text in the figure.

Round 2

Reviewer 2 Report

The authors made modest adjustments to address key assumptions about rat tickling: universality of benefits and generalized operationalization of the technique. Although not the direct remit of the study, I think in the limitations and/or conclusions sections it would be important to speak to the omission in the study design/survey to align welfare measures with implementation. Intent of implementation does not automatically link to value of impact. As scientists and researchers we too often forget this. 

Line 33: add 'potential' to qualify the absoluteness of the assertion made. I.e., “…potential welfare enhancing techniques” This will signpost refinements to assertions in the revised ms.

Line 97: Should be homogeneous not homogenous

Author Response

Reviewer 2 Response

The authors made modest adjustments to address key assumptions about rat tickling: universality of benefits and generalized operationalization of the technique. Although not the direct remit of the study, I think in the limitations and/or conclusions sections it would be important to speak to the omission in the study design/survey to align welfare measures with implementation. Intent of implementation does not automatically link to value of impact. As scientists and researchers we too often forget this. 

  • We have re-formatted and added information to the limitations section to address the limitation of no direct welfare assessment.
    • “This study is not without limitations. First of all, since this study only involved self-report data there is the potential for measurement errors and subjective biases to occur. For example, participants may have inaccurately rated their own implementation, familiarity, and other important outcomes. From experience, we know that some people think that simply scratching a rat on the back of the neck is rat tickling. Therefore, participants – especially at baseline or those assigned to the waitlist groups – may have been unable to accurately rate themselves. Furthermore, trained participants may have become more conservative in their ratings as they became more knowledgeable while waitlist participants over-reported due to the Dunning-Kruger effect where less trained individuals tend to over-report their competence. Thus, if trained participants became more conservative and waitlist participants over-report, then our results may be even stronger than they appear – although we cannot say for sure. This potential measurement error is also a possible explanation for why implementation of rat tickling was not significantly different between groups at the end of this study. However, despite these limitations, our study still provides support for targeted training being beneficial to important outcomes related to rat tickling.
    • Second, although there is strong scientific evidence for the benefits of rat tickling [10], animal-based measures of welfare were not assessed in this study. Therefore, we cannot definitely claim that rat welfare was improved simply because of increased implementation. After all, even trained participants may not correctly apply the technique or self-report their application accurately. Furthermore, even if rat tickling is applied correctly and rated accurately, its application may not ubiquitously increase positive affect and welfare; rats show individual differences in response to tickling and are less responsive when stress or older [10]. However, participants were educated about these factors, taught to evaluate rat responses to tickling, and discontinue if negative responses are continued to be observed. Therefore, if participants followed trained procedures correctly, it is likely – albeit not proven – that rat welfare was improved. Overall, this study still provides valuable insight in the efficacy of training laboratory animal personnel.”

Line 33: add 'potential' to qualify the absoluteness of the assertion made. I.e., “…potential welfare enhancing techniques” This will signpost refinements to assertions in the revised ms.

  • We have added this word to line 33

Line 97: Should be homogeneous not homogenous

  • We have fixed this spelling